# Immunohistochemical Detection of Estrogen Receptor-Beta (ERβ) with PPZ0506 Antibody in Murine Tissue: From Pitfalls to Optimization

**DOI:** 10.3390/biomedicines10123100

**Published:** 2022-12-01

**Authors:** Sarah K. Schröder, Carmen G. Tag, Jan C. Kessel, Per Antonson, Ralf Weiskirchen

**Affiliations:** 1Institute of Molecular Pathobiochemistry, Experimental Gene Therapy and Clinical Chemistry (IFMPEGKC), RWTH University Hospital Aachen, D-52074 Aachen, Germany; 2Department of Biosciences and Nutrition, Karolinska Institutet, Neo, SE 14157 Huddinge, Sweden

**Keywords:** estrogen receptor beta (ERβ), *Esr2*, PPZ0506, immunohistochemistry, staining, antibody validation, ovary, murine tissue

## Abstract

The estrogen receptor beta (ERβ) is physiologically essential for reproductive biology and is implicated in various diseases. However, despite more than 20 years of intensive research on ERβ, there are still uncertainties about its distribution in tissues and cellular expression. Several studies show contrasts between mRNA and protein levels, and the use of knockout strategies revealed that many commercially available antibodies gave false-positive expression results. Recently, a specific monoclonal antibody against human ERβ (PPZ0506) showed cross-reactivity with rodents and was optimized for the detection of rat ERβ. Herein, we established an immunohistochemical detection protocol for ERβ protein in mouse tissue. Staining was optimized on murine ovaries, as granulosa cells are known to strongly express ERβ. The staining results were confirmed by western blot analysis and RT-PCR. To obtain accurate and reliable staining results, different staining conditions were tested in paraffin-embedded tissues. Different pitfalls were encountered in immunohistochemical detection. Strong heat-induced epitope retrieval (HIER) and appropriate antibody dilution were required to visualize specific nuclear expression of ERβ. Finally, the specificity of the antibody was confirmed by using ovaries from *Esr2*-depleted mice. However, in some animals, strong (non-specific) background staining appeared. These signals could not be significantly alleviated with commercially available additional blocking solutions and are most likely due to estrus-dependent expression of endogenous immunoglobulins. In summary, our study showed that the antibody PPZ0506, originally directed against human ERβ, is also suitable for reliable detection of murine ERβ. An established staining protocol mitigated ambiguities regarding the expression and distribution of ERβ in different tissues and will contribute to an improved understanding of its role and functions in murine tissues in the future.

## 1. Introduction

Estrogens play an essential role in development and have key functions in reproductive biology. They mediate cellular signaling by forming a complex with nuclear steroid receptors and interact with specific DNA sequences. Initially, only the existence of one estrogen receptor, estrogen receptor alpha (ERα, encoded by *Esr1*), was described in different species including humans [1], mouse [2] and rat [3]. The murine ERα protein sequence shares high overall homology with rat (97%) and human (88%) estrogen receptors [2] and was found to be widely expressed in many organ systems, including female and male reproductive systems, the central nervous system and the liver [4,5]. In 1996, Kuiper and coworkers described another member of the estrogen receptor family, estrogen receptor beta (ERβ, encoded by *Esr2*), which was primarily localized in the ovary and prostate epithelium of rodents [6,7]. Comparing the distribution of ERα and ERβ, initial studies in rats showed distinct expressions of both receptors in female reproductive organs among the cell types. In uterine tissue, ERα is mainly found in the luminal and glandular epithelial cells. In the ovary, ERβ is detectable in the granulosa cells and ERα is mainly detectable in the interstitial stromal cells and thecal cells [6,8,9]. Using non-radioactive in situ hybridization to detect estrogen receptors, Hishikawa and colleagues localized a similar expression pattern in murine ovaries [10]. In addition, sequential and structural studies revealed that these receptors are conserved via different species [11]. However, studies provided evidence that in some tissues there is high species difference in distribution of ERβ. For instance, human ERβ is highly abundant in testes [12,13,14], whereas murine testes showed virtually no mRNA expression [7].

It is known that human ERα and ERβ share 97% homology in their DNA-binding domain but only 59% in the ligand-binding domain [15]. The generation of different *Esr1* and *Esr2* knockout models provided new insights into the partially divergent functions and expression patterns of these two estrogen receptors in both sexes [16,17,18,19,20,21,22,23,24].

Extensive studies have been performed on the role of estrogen receptors in health and disease. ERβ is involved in various malignancies of the reproductive tract, such as endometriosis, breast, ovarian or prostate cancer [25,26]. Studies showed that not only healthy granulosa cells in the ovary express ERβ, but it is expressed also in cells that had transformed from normal proliferating cells to tumor cells. Granulosa cell tumors are a rare form of ovarian cancer and are characterized by production of steroid hormone receptors and diverse hormones including estradiol [27,28]. Consequently, one therapeutic option for ovarian granulosa cell tumors besides chemotherapy and surgery is the usage of anti-hormonal therapies with, for instance, aromatase inhibitors [29], which however, strongly depend on the hormone receptor status of the tumor [30]. Besides its role in disease of the reproductive tract, ERβ has also multifaceted roles in the pathogenesis of diseases of the non-reproductive system including in esophageal diseases or gastric cancers [25,26]. However, there is still ambiguity regarding the distribution and detection of these, especially in rodent tissues. For both ERα and ERβ, there are a variety of antibodies that bind to different functional domains [14,15,31]. Unfortunately, an increasing number of studies have revealed and disclosed that many of these antibodies give non-specific, false-positive results, or they show relevant discrepancies with *Esr2* mRNA transcripts [14,32]. Remarkably, Andersson and colleagues investigated 13 different antibodies showing that 11 of them failed to reliably detect ERβ in human ERβ-expressing cell lines [14]. From the remaining two, PPZ0506 appeared to be the antibody with the highest specificity to detect ERβ in human cells and tissues [14]. Mouse monoclonal PPZ0506 is designed to bind human ERβ on the *N*-terminal region (2–88 amino acids) and is supposed to detect entire protein including different isoforms [14]. This finding raised the question whether this antibody, due to its high species homology, is also suitable for studies in rodents. Ishii et al. confirmed cross-reactivity of PPZ0506 with rodent ERβ protein [32]. In western blot analysis, transfection of either mouse or human ERβ confirmed specific immunoreactivity of this antibody. In addition, the group validated applicability of PPZ0506 for immunohistochemical detection of ERβ proteins in rat tissues [32] and recently published an optimized protocol using rat ovaries [33].

The application of antibody PPZ0506 in western blot analyses has already been confirmed by others [24,32]. Immunohistochemical staining is essential to differentiate exact localization of ERβ and to distinguish which cells express the receptor. Therefore, the present study was designed to investigate the applicability of the mouse monoclonal antibody PP0Z506 on murine tissues.

## 2. Materials and Methods

### 2.1. Animals

The animals used for this study were acquired and kept in accordance with the recommendations of the Federation of European Laboratory Animal Science Associations. All experiments were approved by the internal Review Board of the RWTH University Hospital Aachen (permit no.: TV40138) and animals were sacrificed without further treatment by cervical dislocation. Male and female wild-type mice (4–5 per cage) with genetic background C57BL/6 were housed in a 12:12 light-dark cycle at constant humidity (50%) and temperature (20 °C) with free access to food and water ad libitum [34]. When the organs were dissected animals were between 6 and 13 weeks old. *Esr2*-depleted mice were generated using CRISPR/Cas9 and housed in Sweden as described elsewhere [24]. Paraffin-embedded tissues or frozen pieces of tissues were sent to Germany for further experiments.

### 2.2. Tissue Preparation

Different reproductive organs and liver tissues were removed from the animals, rinsed in phosphate-buffed saline (PBS) and fixed in 4% neutral buffered formaldehyde (stabilized with methanol) for 24 h. The fixed organs were dehydrated through a graded ethanol series, cleared in xylene, and subsequently embedded in paraffin. Paraffin-embedded tissues were cut in 4 µm thickness. Slices were dried at least for overnight at 37 °C before using for staining. To extract total protein or RNA, organs were removed, rinsed in PBS, immediately frozen in liquid nitrogen, and stored at −80 °C until use.

### 2.3. Hematoxylin-Eosin Staining

Tissue sections were deparaffinized with xylene and decreasing graded ethanol. To obtain an overview of the histology of the tissue, Hematoxylin and Eosin (HE) staining was performed using standard protocols. The estrous phases were determined based on the histology of the female reproductive organs [35,36,37]. After staining and incubation in increasing graded ethanol and xylene, the slices were mounted with DPX Mountant (Sigma-Aldrich, Taufkirchen, Germany) and dried for microscopic assessment. Selected sections were digitized after staining using a NanoZoomer SQ digital slide scanner (C13140-21, Hamamatsu Photonics, Hamamatsu, Japan) and then visualized using the NDP.view 2 software (U12388-01, Hamamatsu Photonics).

### 2.4. Immunohistochemistry Staining Protocol

For immunohistochemistry detection of ERβ, 4-µm-thick sections of mouse tissues were deparaffinized with xylene and decreasing graded ethanol. Antigen retrieval was performed by heating the sections in sodium citrate buffer (10 mM, 0.05% Tween 20, pH 6.0) in a steamer of 30 min, followed by cooling on ice for 20 min. Subsequently, the sections were briefly immersed in PBS and then in PBS with 0.1% Tween 20 (PBS-T). Tissue was subjected to Biotin Blocking System (#X0590, Dako Agilent Technologies, Inc., Santa Clara, CA, USA), which consists of Avidin and a Biotin solution. Each solution was incubated for 15 min, while in-between washing steps were performed with PBS and PBS-T. This and all subsequent incubation steps were performed in a moist chamber, while washing was done in glass cuvettes. Unspecific-binding sites were blocked by incubation in 5% normal goat serum (#X0907, Dako Agilent Technologies) in blocking solution (1% BSA, 0.1% cold fish gelatin, 0.1% Triton-X-100, 0.05% Tween 20 in PBS) for 90 min. When indicated, VisUBlock Mouse on Mouse Blocking Reagent (VB001-01ML, R & D Systems, Inc., Minneapolis, MN, USA) was used in some sections according to the manufacturer’s instructions. Next, slices were incubated with different concentrations of anti-ESR2 (#PP-PPZ0506-00, R & D Systems) primary antibody diluted in blocking solution overnight (4 °C). As a negative control, tissue sections were incubated with a mouse IgG_2b_ isotype control (#MAB004, R & D Systems), where the final antibody concentration was equal to isotype control concentration. The next day, quenching of endogenous peroxidase was performed and sections were incubated in 3% hydrogen peroxide (Sigma-Aldrich) in methanol for 15 min. Tissue sections were washed in dH_2_O and PBS-T, followed by an incubation with biotinylated polyclonal goat anti-mouse secondary antibody (#E04331, Dako Agilent Technologies), diluted 1:200 in PBS for 1 h. After washing the sections three times in PBS-T, tissues were covered with ABC-Complex solution (#PK-6100, VECTASTAIN^®^ Elite^®^ ABC-HRP Kit, Vector Laboratories Inc, Newark, CA, USA) according to the manufacturer’s instructions for 1 h. To detect ERβ expression, Vector NovaRED^®^ Substrat Kit for peroxidase was used (#SK-4800, Vector Laboratories), which produces a red reaction product. After incubation for 5 min, slices were immersed in dH_2_O and washed in PBS. Next, tissue sections were incubated with Methyl Green Counterstain (#H-3402, Vector Laboratories) in a water bath preheated to 60 °C for 5 min. The samples were rinsed with tap water and immediately subjected to increasing graded ethanol. After final steps in xylene the slices were mounted in DPX Mountant (#06522, Sigma-Aldrich) and dried for microscopic analysis. Selected sections were digitized after staining using a NanoZoomer SQ digital slide scanner as described above.

### 2.5. RNA Extraction and RT-PCR

For RNA extraction, snap-frozen tissues were homogenized in RNA lysis buffer (containing 40 mM dithiothreitol) in a Mixer Mill MM 400 homogenizer (Retsch GmbH, Haan, Germany) using 2 sterile grinding balls (#22.455.0002, Retsch GmbH) for 5 min (30 Hz). After removal of grinding balls, samples were centrifuged for 6 min at 3000× *g*. The supernatant was used for purifying total RNA with PureLink RNA mini kit including DNase digestion according to the manufacturer’s guidelines. cDNA was synthetized from 1 µg RNA using SuperScript II reverse transcriptase (#18064-022, Thermo Fisher Scientific Inc., Waltham, MA, USA) as described previously [38]. For conventional RT-PCR, cDNA was subjected to the following cycle conditions: 5 min 95 °C, followed by repeated cycles of 1 min 95 °C, 1 min at 60 °C (*Actb* #20 cycles, *Esr2* #35 cycles, *Rn18s* #25 cycles), 3 min 72 °C and a final temperature step at 72 °C for 10 min. The murine primers (*Actb,* NM_007393.5: for: 5′-CTC TAG ACT TCG AGC AGG AGA TGG-3′, rev: 5′-ATG CCA CAG GAT TCC ATA CCC AAG A-3′, 163 bp; *Esr2*, NM_207707.1: for: 5′-GAC GAA GAG TGC TGT CCC AA-3′, rev: 5′-TCA GCT TCC GGC TAC TCT CT-3‘, 209 bp; *Rn18s,* NR_003278: for: 5′-CTC AAC ACG GGA AAC CTC AC-3′, rev: 5′-CGC TCC ACC AAC TAA GAA CG-3′, 110 bp) were spanning exon junctions and were purchased from Eurofins Genomics Germany GmbH (Ebersberg, Germany). Amplicons were separated in 1.6% agarose gels with ethidium bromide using 1× TAE (40 mM Tris base, 20 mM acetic acid, and 1 mM ethylenediaminetetraacetic acid disodium salt dihydrate) running buffer and visualized in a GEL iX20 imager (INTAS Science Imaging Instruments GmbH, Göttingen, Germany).

### 2.6. Protein Extraction and Western Blot Analysis

For protein analysis, pieces of respective tissues were immediately collected in RIPA buffer (50 mM Tris-HCl (pH 7.2), 150 nM NaCl, 1% (*w*/*v*) NP-40, 0.1% (*w*/*v*) SDS, 0.5% (*w*/*v*) sodium deoxycholate) containing cOmplete™-proteinase inhibitor (#11697498001, Merck KGa, Darmstadt, Germany) and phosphatase inhibitor cocktail II (#P5726, Sigma-Aldrich). For homogenization of the tissue, 2 sterile stainless steel grinding balls were placed per tube and tubes were transferred in the homogenizer as described above for RNA extraction. After homogenization, grinding balls were carefully removed and samples were centrifuged at 10,000× *g* for 15 min (4 °C) to separate proteins (supernatant) from debris. To remove fat residues, the centrifugation step was repeated 3 times. The protein amount was determined by the DC protein assay (#500-0116, Bio-Rad Laboratories GmbH, Düsseldorf, Germany). Equal amounts of proteins (40 µg) were mixed with dithiothreitol as a reducing agent and Nu-PAGE™ LDS electrophoresis sample buffer (#NP0008, Thermo Fisher Scientific). Before separating proteins in 4–12% Bis-Tris gradient gels (#12020166, Invitrogen, Waltham, MA, USA) using 3-(*N*-morpholino)propanesulfonic acid (MOPS, 1×) running buffer, samples were heated at 80 °C for 10 min. The proteins were blotted on nitrocellulose membrane (#GE10600001, 0.2 μm, Merck, Darmstadt, Germany) and successful transfer was confirmed by Ponceau S stain. Non-specific binding sites were blocked with 5% (*w*/*v*) non-fat milk powder in Tris-buffered saline with Tween 20 (TBST). The membranes were incubated with primary antibodies specific for ERβ (1:2000, #PP-PZ0506-00; R & D Systems) and β-actin (1:10,000, #A5441, Sigma-Aldrich) overnight (shaking, 4 °C) and subsequently visualized with secondary antibodies coupled to horseradish peroxidase with SuperSignal chemiluminescence substrate (#34076, Thermo Fisher Scientific).

### 2.7. Aligment of ERβ Protein Sequence in Different Species

To compare the sequence homology between the human ERβ receptor (hERβ) and rodent ERβ, the protein sequences of the different species were aligned via the freely available Basic Local Alignment Search Tool (BLAST) software [39]. The following protein accession numbers were used for analysis: hERβ P03372.2, mERβ O08537.3 and rERβ Q62986.2, respectively. Sequence identity was calculated over the entire length of the protein (530 residues), as well as homology in the region of binding of the PPZ0506 antibody (2–88 amino acids).

## 3. Results

### 3.1. Detection of ERβ in Murine Tissues

Although estrogen receptor 2 (ERβ) was discovered more than 25 years ago, there are still discrepancies about its expression and distribution in various tissues [6,14,40]. Despite a variety of existing antibodies for the detection of ERβ, several studies revealed false-positive results using *Esr2*-knockout or transfection strategies [14,40,41]. Recently, it was demonstrated that the anti-human ERβ antibody PPZ0506 is cross-reactive for rodent ERβ protein [24,32].

The alignment of protein sequences of hERβ with rodent ERβ, performed in this study, confirms a high probability of cross-reactivity. The results of the BLAST analysis show that compared to hERβ, mERβ has a homology of 88.68% and rERβ has a homology of 88.60% along the entire length of the protein (Appendix A). At the *N*-terminal region, where the PPZ0506 antibody binds to the hERβ (between 2–88 AA), hERβ has 83.91% sequence identity to the mERβ and 80.46% to the rERβ.

By using different reproductive organs, and the liver as a non-reproductive tissue, we first examined whether the PPZ0506 antibody can be used to detect ERβ in western blot analysis. Our data show strong expression of ERβ in the murine ovary, whereas uterus and testis were obviously negative (Figure 1A). However, when the exposure time was extended, several non-specific bands lower than 51 kDa appeared in different tissues (indicated by asterisks). That these are specific bands was ruled out by using *Esr2*-deficient animals, as they are as well present in *Esr2* knockout tissues (see Section 3.3 and Appendix A). In addition, neither female nor male liver tissue showed expression of ERβ protein. This was confirmed in mRNA level by RT-PCR, as the 209 bp amplicon for *Esr2* was detected only in the murine ovary (Figure 1B).

Contrary to western blot analysis, immunohistochemical staining can differentiate which cell types express ERβ. When we established the ERβ staining procedure for murine tissues with PPZ0506, there was no optimized protocol available. Our results suggested that strong expression of ERβ is preferentially detected in the ovary. Using the optimized staining protocol that we established for PPZ0506, we were able to confirm this, as ERβ-positive cells were evident in the ovary (Appendix A and Figure 1C). In addition, routine Hematoxylin-Eosin (HE) staining was performed to identify the individual cell types in the murine ovary (Figure 1D). In the optimized staining result, the nuclear ERβ-positive granulosa cells stood out from the other sparsely and weakly stained cells and the non-specific background (Figure 1C,D). Finally, we confirmed the results from western blot and RT-PCR by using the optimized staining protocol in other tissue types showing that in the uterus and testis, as well as in the liver, the staining with PPZ0506 showed no ERβ-positive cells (Appendix A–D).

On the way to an optimized and valid staining protocol, many difficulties and hurdles appeared which will be represented in the following sections of this publication. The herein presented findings should serve as a solid guide for scientists to reliably identify ERβ in their murine tissues and explore its distribution.

### 3.2. Antigen Retrieval Is Mandatory for Nuclear ERβ Detection

It is well known that most of the formaldehyde-fixed tissues require proper antigen retrieval to unmask antigen sites prior immunohistochemical staining [42,43,44]. To investigate whether antigen retrieval is required for detection of Esr2 with the PPZ0506 antibody, ovarian tissues were either incubated in sodium citrate buffer (10 mM, 0.05% Tween 20, pH 6.0) at 100 °C in a steamer for 30 min or subjected to staining without this unmasking step. Interestingly, samples without heat-induced epitope retrieval (HIER) showed only unspecific background staining. Red reaction product using the chromogenic NovaRED^TM^ peroxidase system accumulated between the nuclei of the granulosa cells and in interstitial cells of the ovary (Figure 2A). Preliminary experiments showed that the use of NovaRED^TM^ with methyl green counterstaining gave more precise contrast compared with the often-used 3,3′-diaminobenzidine peroxidase substrate (brown color) with hematoxylin counterstaining when detecting expression of nuclear proteins. Expression of ERβ in uterus is controversial, but studies with validated antibodies suggest that ERβ is not detectable in this tissue [24,33]. Herein, we observed intense red (non-nuclear) unspecific staining in the myometrium but not in the epithelial cell layer (Figure 2B) in murine uterine tissue.

Specific nuclear localization of ERβ protein was detected by antigen retrieval. The nuclei of the granulosa cells in the murine ovary were strongly positive for ERβ (Figure 2A, middle), whereas the majority of the other cells were stained by methyl green. In the uterus, antigen retrieval reduces unspecific background staining, as only a few areas in the myometrium displayed slight red color.

We observed that HIER led to a weaker counterstaining with methyl green. Therefore, the incubation time for methyl green was adjusted from initially 1 to 5 min for all subsequent experiments. As only with antigen retrieval a specific nuclear ERβ signal could be detected, HIER step was included in all subsequent stainings.

### 3.3. Immunhistochemical Detection of Nucelar ERβ

In some cases, it is difficult to distinguish between patterns of non-specific and specific staining when localizing the protein of interest. Therefore, it is critical to include negative controls during the staining procedure. Herein, we used isotype-specific immunoglobulin (mouse monoclonal IgG_2b_) as a negative control in the same concentration as the primary antibody to distinguish non-specific from specific staining results. This control confirms the dependability of the antibody, as the strongly ERβ-positive granulosa cells stained with PPZ0506 remain unstained when the same sections were incubated with IgG_2b_ instead (Appendix A). In addition, the uterine tissue showed no specific staining with the antibody or with the IgG_2b_ control (Appendix A).

For profound antibody validation in an application-specific manner, it is essential to ensure that the antibody is specific, reliable and has high reproducibility [45]. A powerful tool in validation of an antibody for immunohistochemistry is the genetic manipulation of the target of interest [46]. For this purpose, we tested the monoclonal PPZ0506 antibody in tissues from *Esr2*-deficient mice. These animals were generated using the CRISPR/Cas9 technology, where the entire *Esr2* gene was deleted [24]. Ovaries of *Esr2* knockout (*Esr2*^−/−^) animals and wild-type (*Esr2*^+/+^) animals from the same mouse strain were stained for ERβ using PPZ0506. Only the wild-type *Esr2*^+/+^ ovaries showed strong nuclear staining in the granulosa cells, whereas the ovaries of *Esr2*^−/−^ animals did not show specific staining in the nuclei (Figure 3A,B). Similarly, there was no specific nuclear signal in the respective IgG_2b_ negative controls. However, a slight brownish unspecific coloration of the tissue was observed in both genotypes. Expression of ERβ using PPZ0506 was confirmed on protein level by western blot analyses and on mRNA level using conventional RT-PCR. Except in the wild-type ovaries, neither uterus nor testis expressed *Esr2* mRNA or protein (Appendix A).

In conclusion, the use of *Esr2*-knockout ovaries confirmed the reliability and specificity of the mouse monoclonal antibody PPZ0506 for immunohistochemical detection of mouse ERβ protein.

### 3.4. Titration of PPZ0506 Primary Antibody

To obtain an optimal staining result, the antibody used should be validated in a dilution series. According to the well-established protocol and distribution criteria of ERβ in rat ovaries [32], the monoclonal antibody PPZ0506 was tested in murine ovaries at dilutions ranging from 1:250 to 1:16,000 (Figure 4A). Uterine tissue, negative for ERβ protein, was tested as well (Figure 4B). In murine ovaries, strongly and consistently stained nuclei of granulosa cells were observed at dilutions 1:250 to 1:8000. The oocyte cells showed strong staining between 1:250 and 1:4000. Starting at 1:6000 dilutions, the oocyte was only faintly stained or not stained at all. At this and higher dilutions, ERβ was rarely detected in theca cells, while the previously strong non-specific background staining could be minimized. Excessive dilutions (1:16,000) of PPZ0506 antibody still stained some nuclei of granulosa cells, but the signals were inconsistent and blurred. In uterine tissue, strong unspecific staining of the glandular epithelium and the stroma was seen when using PPZ0506 in dilution of 1:250–1000 (Figure 4B). These off-target signals were reduced by diluting the antibody 1:4000 and completely abolished at 1:6000 and above.

In summary, ERβ can be precisely detected in immunohistochemistry staining at a dilution of 1:6000–1:8000 with antibody PPZ0506 in formaldehyde-fixed, paraffin-embedded mouse ovaries, whereas ERβ-negative (uterine) tissues showed no specific staining.

### 3.5. Non-Specific Background Staining

The development of monoclonal antibodies revolutionized the work of scientists and increased the use of immunized mice for hybridoma technology to produce antibodies [47,48]. However, when using primary mouse IgGs on mouse tissue, a well-known phenomenon is the ‘mouse-on-mouse’ problem. This occurs because the secondary antibody cannot distinguish between endogenous mouse immunoglobulins (Igs) in the tissue and mouse primary antibody, leading to false-positive staining results [49]. In the present study, tissues from mice at different estrous stages were used. Strikingly, in some animals intense staining was visible in the ERβ-negative uterine tissue (Figure 5A) and in others not (Figure 5B). It is important to note that this signal was not specific for ERβ, as the same areas were stained when normal IgG_2b_ was used instead of the primary antibody PPZ0506. In addition, the sections were stained by omitting the primary antibody. Besides, another secondary antibody system (i.e., rabbit anti-mouse instead of goat anti-mouse) was also tested, which showed a similar outcome. Most likely, these (non-specific) background signals are due to the presence of endogenous immunoglobulins that fluctuate between estrous stages of animals [50].

After the staining of ERβ in the ovary was established, we next aimed to reduce this previously described non-specific background in uterine tissue. Several manufacturers developed commercially available blocking kits. Herein, we tested the VisUBlock^TM^ Mouse on Mouse Blocking Reagent recommended to effectively block endogenous IgG in mouse tissue. Unfortunately, uterine tissue sections showed higher background signals when using the VisUBlock blocking reagent (Figure 6).

In summary, the mouse monoclonal PPZ0506 antibody is suitable to localize ERβ protein expression specifically and reliably in murine ovaries. Nevertheless, when staining other organs, such as uterine tissues, intense background staining might occur.

## 4. Discussion

Nowadays, there are many commercially antibodies available for nearly all proteins. However, at the same time there are reports showing that many antibodies have insufficient specificity, sensitivity and lot-to-lot consistency that provoke irreproducible and results [51,52,53]. Prototypically, it has been found that many estrogen-receptor beta (ERβ) antibodies are of poor quality and produce false-positive signals [31,40,41]. The anti-human mouse monoclonal antibody PPZ0506 shed light in the controversial studies regarding ERβ expression, as it was tested in human cell lines [40] and clinical material [14] to specifically recognize human ERβ. Recently, others found cross-reactivity of this antibody to rodent tissue [24,32,33].

ERβ is involved in various diseases and different cancer types of the reproductive tract, as well as in other non-reproductive diseases, such as malignancies of the gastrointestinal tract [25,26]. Although not necessarily transferable to humans, mouse knockout models are a valuable tool to clarify the role of ERβ in health and disease [23]. Therefore, it is essential to establish antibodies that accurately and specifically recognize ERβ in murine tissues. To demonstrate that the antibody PPZ0506 has the potency to meet these requirements, we aligned the protein sequences of human and rodent ERβ. We found nearly 89% similarity between human and mouse ERβ along the entire length of the protein and 84% in the antibody binding region, indicating a high probability of species cross-reactivity. In support of our study, a recent study showed that the antibody has the ability to detect ERβ in mouse tissues using western blot analysis [24].

The aim of the present study was to establish an immunohistochemical staining protocol for the detection of ERβ in mouse tissue. This tool would provide the opportunity to specifically identify the cells that express ERβ, which is essential for further research questions. The results obtained with the optimized staining procedure on various murine tissues are consistent with western blot analysis and RT-PCR. Similar to the establishment of PPZ0506 on rat tissue [32], we found that many variables within the optimization process have an impact on the staining result.

In the following, we would like to discuss which steps in the present protocol have been validated and are essential to achieve solid staining results. Estrogen receptors are predominantly localized in the nucleus as they hold a constitutive active nuclear localization signal [54]. As one essential step in the optimized procedure, we found that besides permeabilization with Triton-X-100 (0.1% as part of blocking solution), a strong heat-induced antigen retrieval (HIER) is required to detect any specific nuclear signal for the ERβ in the granulosa cells of the ovary. On the contrary, slices incubated without HIER showed diffuse unspecific cytoplasmic staining. Similar findings were obtained by Hattori and colleagues, who reported that ERβ staining in rat tissues without HIER give rise to high background signals [33].

It is known that the preservation of tissues for immunohistochemical staining using formalin can result in intramolecular cross-links between the fixative and other proteins, masking the linear epitopes for accurate antigen-antibody recognition [55]. In addition, PPZ0506 is designed to bind on the *N*-terminal region of ERβ, which represents a site for post-translational modifications and possible interactions with transcription factors [56,57]. Since this could possibly influence or even prevent binding of the antibody to the protein, this circumstance is discussed as one of the possible reasons for the controversial studies with various ERβ antibodies [33,56]. Furthermore, there are reports demonstrating that in some cases multiple antigen retrieval methods are required to obtain an adequate staining signal [58]. Our data show that the HIER step included in the present protocol is sufficient for proper detection of ERβ with PPZ0506. We confirmed once again that HIER is a powerful method to improve the accessibility in immunohistochemical stainings, particularly when dealing with nuclear targets [42,43]. Nevertheless, the exact mechanisms by which the HIER procedure works is not yet fully described. It is most likely that the different variations of this methodology provoke hydrolytic cleavage of formaldehyde-related chemical groups and crosslinks, unmasking of inner epitopes, and the extraction of calcium ions from protein complexes that decrease the electrostatic interactions between protein networks [44].

In order to obtain a specific staining result, beside the selection of the convenient antibody, also the choice of the appropriate controls is essential. A profound misconception in antibody validation is to use negative controls that omit the primary antibody. Such staining results show only the non-specific binding of a secondary antibody, but not of the primary antibody [59]. In the present study, an isotype-specific mouse monoclonal immunoglobulin (IgG_2b_) was chosen as the negative control instead of primary antibody PPZ0506. The obtained results underline the specificity of the antibody, which specifically stained ERβ-positive granulosa cells in the ovary that remained unstained in negative control slices.

One of the pillars for a comprehensive validation of an antibody proposed by Uhlen and coworkers also includes a genetic approach [45]. It is recommended to use knockdown or even knockout techniques, such as CRISPR/Cas9, to discriminate between specific and non-specific staining. Herein, we used tissues from an all-exon *Esr2*-depleted mouse line that was genetically edited by CRISPR/Cas9 strategy [24]. RT-PCR and western blot analysis confirmed the complete elimination of *Esr2* and ERβ protein level in murine ovaries. When ovaries of *Esr2*-deficient (*Esr2*^−/−^) and wild-type (*Esr2*^+/+^) mice were comparatively stained with PPZ0506 or IgG_2b_, only the granulosa cells of wild-type animals were stained, again showing the specificity of the PPZ0506 antibody. The genetic approach emphasizes the applicability of the PPZ0506 antibody on murine tissues, as it showed no specific staining in tissue lacking ERβ.

Another essential optimization step is the titration of the antibody, which enables the discrimination between positive from non-specific background staining [60]. Since the staining outcome depends on a plenty of different factors [58], the advice is to start with a working concentration of 1 to 5 µg/mL of the primary antibody. In our study, the antibody PPZ0506 showed strong background signals at lower dilution (1:250 corresponding to a concentration of 4 µg/mL) in both ovarian and especially ERβ-negative uterine tissues. The fact that specific ERβ-positive nuclei can be detected even when a strong non-specific cytoplasmic background is present has already been shown in mammary gland tumor cells [60]. With higher dilution (1:16,000 corresponding to a concentration of 0.06 µg/mL), we observed that the specific signal in the granulosa cells of the ovary slowly faded. Depending on the batch and staining conditions, a specific signal was obtained when the antibody was used in the range of 1:4000 to 1:8000 (corresponding to 0.25 to 0.125 µg/mL). This is in line previous findings showing that antigen-retrieval lowers the working concentration of the antibody [58].

However, it is known that in some areas or cell types such as immune cells (supposed to lack the target of interest), signals occur even when the antibody was strongly diluted. In such cases, not only the antibody-stained sections, but also the control sections (IgG stained) show intense non-specific staining [60,61,62]. These findings are consistent with our results showing partial strong non-specific signals in the (ERβ-negative) uterus in primary antibody-stained sections and in controls. In addition, these unwanted side-effects are elevated when using mouse primary antibodies on murine tissues, which were previously described as ‘mouse-on-mouse’ problem [49,58]. Since the secondary antibody cannot differentiate between endogenous mouse Igs and the primary antibody, these effects might occur. The presence of endogenous Igs, especially in intracellular space or blood vesicles, is known to vary from tissue to tissue [60,62]. In the present study, the murine ovary showed moderate non-specific background in contrast to uterine tissue with very strong signals in some samples. However, including a step with a commercially available blocking solutions for the ‘mouse-on-mouse’ problem (VisUBlock, as a potential reagent), did not contribute to a reduction of the non-specific background in our study.

In parallel with our study, another research group has recently worked on the improvement of immunohistochemical detection protocols for ERβ in murine tissues [63]. The respective research group came up with a protocol that methodologically differs from our protocol in some steps, such as the use of HRP-polymer-bound secondary antibodies instead of the avidin-biotin detection system [63]. In line with our findings, the authors concluded that the monoclonal antibody PPZ0506 directed against amino acids 2–88 of human ERβ is a suitable mean to detect ERβ in murine tissues. Moreover, to confirm ERβ expression, the authors performed an RNAScope in situ hybridization assay, representing an excellent tool for visualizing mRNA localization in tissues. Interestingly, the authors present an additional step that can be applied to further reduce non-specific background effects. They show that one way to reduce endogenous Igs is to perfuse the living animals in deep anesthesia with saline buffer and fixative before tissue collection [63]. To perform such technique, an application requiring approval is a mandatory in the European Union according to Directive 2010/63/EU [64]. In order to provide an optimized staining protocol to a broad research community at feasible conditions that are in line with 3Rs principle (i.e., Replacement, Reduction, Refinement) [65], we decided to work with organs from already-sacrificed animals.

When working with female animals, one should not neglect the influence of estrus on the experimental setup. Particularly, tissues are affected, which are naturally subject to strong hormonal fluctuations and histological changes, such as the uterus. Since the amount of Igs is significantly altered in different estrus stages, stage-dependent variability of background staining can be expected [50]. It is well known that animals in proestrus have the highest amount of Igs especially in the stromal cells [50], what is in line with our results. In addition, high variations in ERβ staining intensity can occur in rat ovaries depending on estrus stage [32]. Our study confirms this notion, because we observed fluctuations in staining intensity in the ovaries at different stages. It is thought that natural hormonal fluctuations may affect the sensing of certain targets. This phenomenon was previously discussed in the diagnostic determination of ERβ expression in normal and malignant breast tissue [66].

In sum, all these factors discussed can influence the staining results and making specific protein detection difficult in some circumstances. Therefore, when working with females, precise determination of estrus stage is essential and comparing animals at one stage recommended. If possible, tissues from animals harvested in the diestrus phase should be ideally used [24,32,33,63]. The possibility of using the same antibody (PPZ0506) in different species brings clarity to the controversial studies of the past years and shows precise interspecies differences in localization of ERβ [63]. Finally, the presented data show that hormonal influences can affect immunohistochemical staining results and non-specific background makes it difficult to delineate positive staining. The clinical application of ERβ diagnosis in tissues should also take into account the potential influence of natural hormonal fluctuations, e.g., during menstrual phases or in menopausal patients. In addition, the protocol provided will increase the biological significance of experiments that are conducted in mouse since proper staining procedures for individual components of the estrogen signaling pathway are indispensable to establish valid expression data in translational research.

## 5. Conclusions

In conclusion, based on the data presented, mouse monoclonal antibody PPZ0506 is suitable for the specific detection of ERβ in murine tissues. The use of blocking steps and strong heat-induced antigen retrieval enabled a reliable and reproducible staining protocol. The use of IgG_2b_ control and *Esr2*-deficient animals confirmed that the granulosa cells in the murine ovary are strongly positive for ERβ. In addition, these staining results are supported by western blot analysis and RT-PCR. Nevertheless, depending on the tissue type, unspecific background signals might occur when using the mouse monoclonal antibody PPZ0506, which can only be reduced to a limited extent. However, after precise titration of the primary antibody and by using appropriate controls, PPZ0506 is highly recommended for localization of ERβ not only in human, but also in rat and mouse, tissues.

## Figures and Tables

**Figure 1 biomedicines-10-03100-f001:**
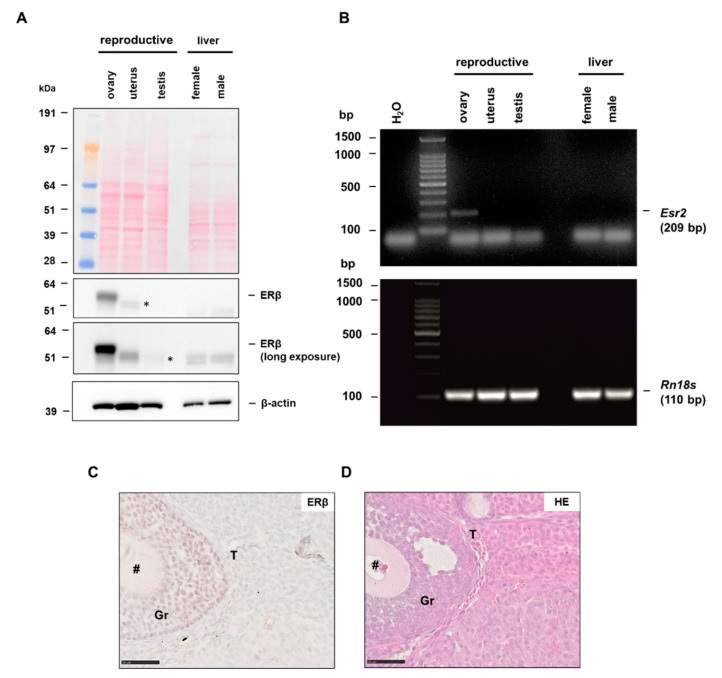
Estrogen receptor beta (ERβ) expression in different tissues. Murine tissues were collected and processed for protein or mRNA analysis as described in Material and Methods: (**A**) ERβ protein expression was analyzed by western blot analysis in ovarian, uterine, testis and liver tissue extracts with monoclonal antibody PPZ0506. Ponceau S stain confirms successful transfer and β-actin re-probing served as an internal loading control. ERβ protein could only be detected in the murine ovary. Asterisks (*) indicate non-specific bands: (**B**) in line, the 206 bp *Esr2* amplicon amplified by RT-PCR in ovarian tissue was absent in all other analyzed tissues. The product of *Rn18s* (110 bp) served as a control; (**C**) immunohistochemical staining for ERβ using PPZ0506 in optimized conditions show strong ERβ-positive granulosa cells (Gr). Further cell types indicated are: #, oocyte cell, T, theca cells; and (**D**) routine HE staining of murine ovary showed the general histology of the tissue. Scale bar 50 µm (400×).

**Figure 2 biomedicines-10-03100-f002:**
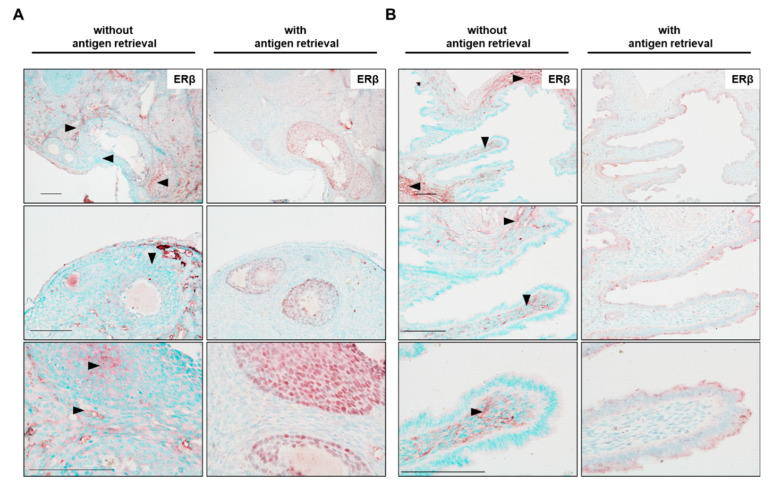
Heat-induced antigen retrieval is mandatory to obtain specific nuclear estrogen receptor beta (ERβ) staining. Murine ovaries (**A**) or uterine tissue (**B**) were used to evaluate whether antigen unmasking is required for specific nuclear ERβ staining using PPZ0506 antibody. Some sections were stained without antigen retrieval, whereas in others antigen retrieval was performed by heating the sections in sodium citrate buffer (10 mM, 0.05% Tween 20, pH 6.0) in a steamer for 30 min. In the depicted experiment, PPZ0506 antibody was diluted 1:6000. Please note that slides without antigen retrieval showed no nuclear staining at all whereas unspecific cytoplasmatic staining occurs (arrowheads). The used antigen retrieval method enabled a dependable and specific nuclear ERβ staining (of the granulosa cells) in the ovary, but no specific staining in the uterus. Scale bars, 100 µm (in all magnifications).

**Figure 3 biomedicines-10-03100-f003:**
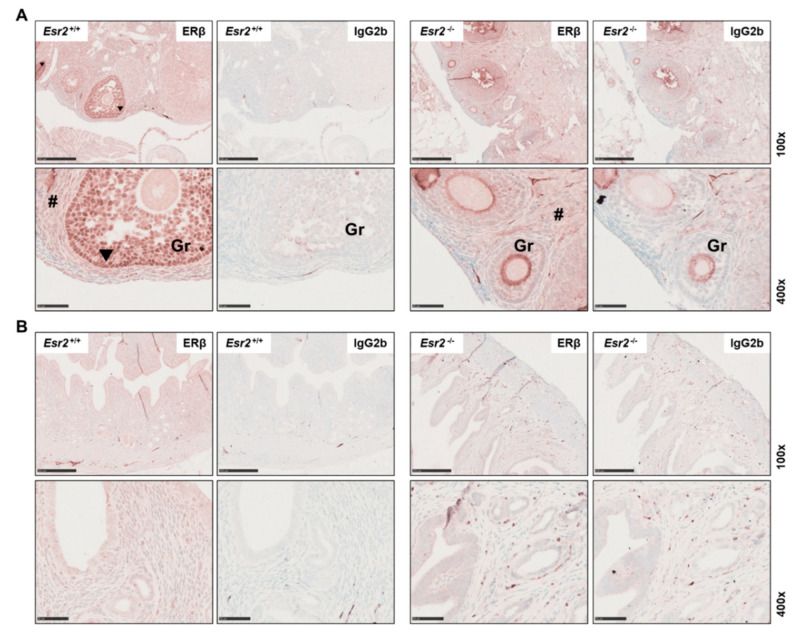
Female reproductive tissues of *Esr2*-depleted mice are negative for estrogen receptor beta (ERβ) expression. Ovaries (**A**) and uterine tissue (**B**) of *Esr2* knockout (*Esr2*^−/−^) and wild-type (*Esr2*^+/+^) animals from the same mouse line were stained with PPZ0506 antibody or with isotype-specific IgG_2b_ negative control. Only granulosa cells (Gr) in *Esr2*^+/+^ ovaries strongly express ERβ, whereas in *Esr2*^−/−^ ovary and uterine tissue no specific nuclear staining was observed. Specific staining is shown by arrowheads in *Esr2*^+/+^ and # indicates diffuse non-specific background staining. Scale bar 250 µm (100×) or 50 µm (400×).

**Figure 4 biomedicines-10-03100-f004:**
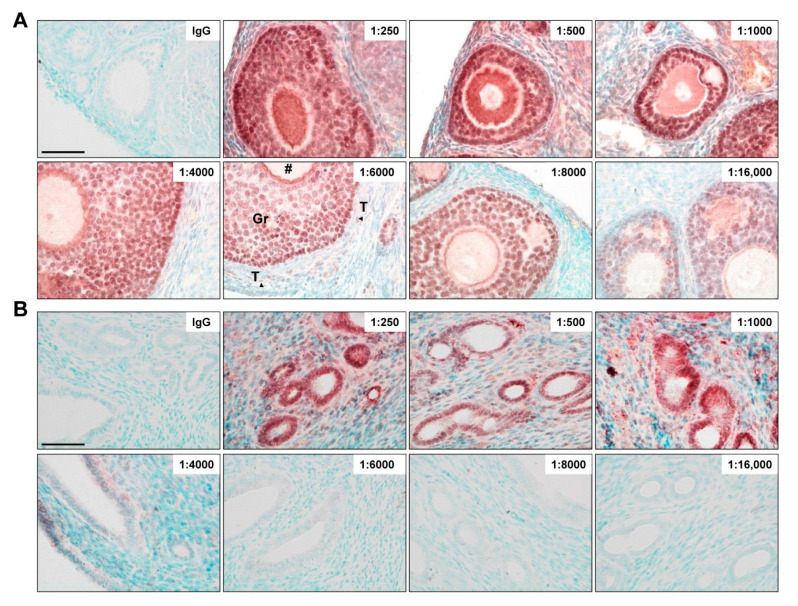
Titration of PPZ0506 primary antibody. Paraffin-embedded sections were chosen for immunohistochemical detection of nuclear expressed estrogen receptor beta (ERβ). The antibody (PPZ0506; 1 mg/mL) was diluted to obtain optimal dilution ratio (as indicated) in ovaries (**A**) or uterus (**B**). For negative control, IgG isotope staining was performed with a concentration equivalent to primary antibody dilution (1:8000). Cell types of the ovary are indicated as: Gr, granulosa cells; T (arrowheads), theca cells; #, oocyte cells. Please note that the murine uterus tissue was negative for ERβ. Scale bar 50 µm (400×).

**Figure 5 biomedicines-10-03100-f005:**
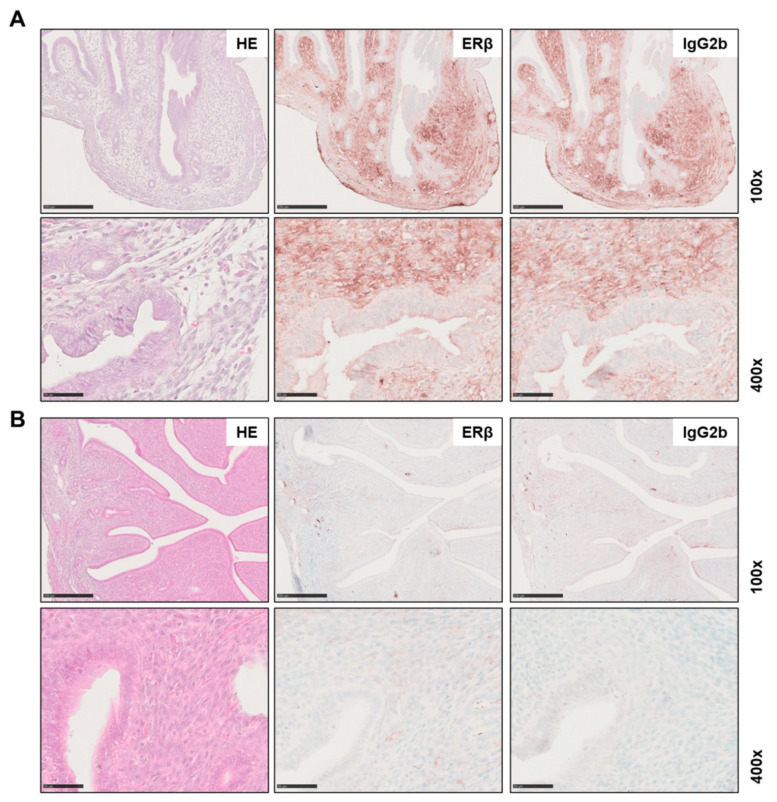
Non-specific background signal in uterine tissue. Tissue slices were prepared as described in Material and Methods for routine HE stains or immunohistochemical detection of estrogen receptor beta (ERβ) using PPZ0506 or IgG_2b_ isotope control. Uterine tissue form mice in (**A**) the proestrus and (**B**) the metestrus stage of the estrous cycle are shown. Please note that there are strong differences in non-specific background staining intensity between the different stages. Scale bar 250 µm (100×) or 50 µm (400×).

**Figure 6 biomedicines-10-03100-f006:**
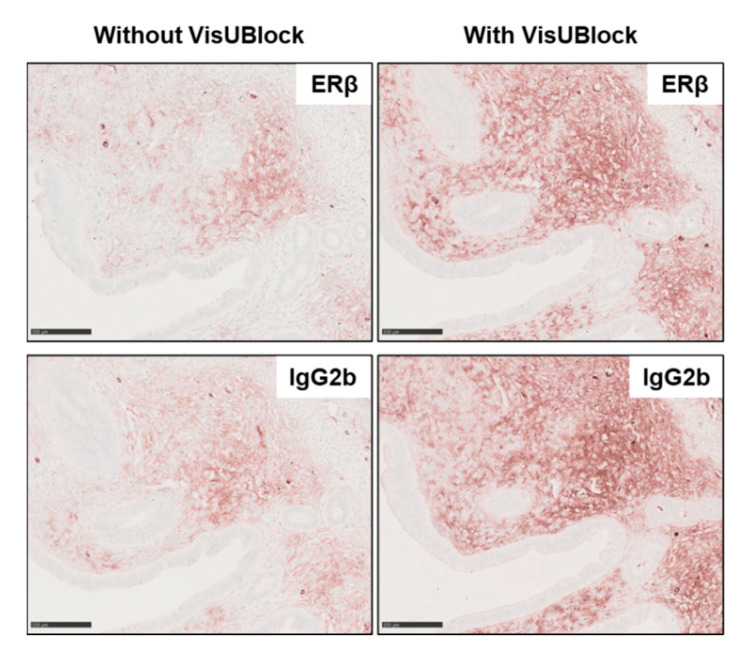
Effects of VisUBlock in immunohistochemical detection. Tissue slices of murine uterine tissue were prepared as described in Material and Methods including incubation for 1 hour with VisUBlock blocking reagent prior staining with PPZ0506 antibody or isotype-specific IgG_2b_ negative control. The use of VisUBlock increased unspecific background signals in the uterine tissue. Scale bar: 100 µm (200×).

## Data Availability

The original data of this study is stored in the Institute of Molecular Pathobiochemistry, Experimental Gene Therapy and Clinical Chemistry (IFMPEGKC) located at the RWTH University Hospital Aachen. Results of repetitions that are not shown here can be requested from the corresponding author.

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
