# Peer review of "Immunohistochemical Detection of Estrogen Receptor-Beta (ERβ) with PPZ0506 Antibody in Murine Tissue: From Pitfalls to Optimization"

_biomedicines, 2022, doi:10.3390/biomedicines10123100_

Round 1
Reviewer 1 Report
The article is very interesting. The research methodology and results were meticulously presented by the Authors. Thanks to this, the article has great potential for the instructions provided to be used in future research by other scientists.
A month ago, an article (doi: 10.1267/ahc.22-00043) on similar issues was published. Can the methodology and results of this study also be included in your discussion?
Author Response
Dear reviewer 1,
many thanks for reviewing our paper. Please find attached a pdf-File in which we describe how we dealt with your concerns/suggestions.
Regards
Ralf Weiskirchen

Reviewer 2 Report
In this manuscript the authors present an immunohistochemical staining protocol to detect ERβ in mouse tissue. It is a well written manuscript with clearly presented methodology and results. I have the following comments:
1. In which diseases has the role of ERβ been established?
2. ERβ is expressed in the granulosa cells of the ovary. What is the role of hormonal treatment in granulosa cell tumours of the ovary? A short comment should be added to the introduction section.
3. In the discussion section the authors should further discuss possible clinical implications of these findings. In what way do these findings have a potential to improve clinical practice?
Author Response
Dear reviewer 2,
many thanks for reviewing our paper. Please find attached a pdf-File in which we describe how we dealt with your concerns/suggestions.
Regards
Ralf Weiskirchen

Reviewer 3 Report
The authors have done a wonderful job describing this novel technique of immunohistochemical detection of ER. I do not find a part of the study that required major modification; hence, i believe it can published in its present form.
Author Response
Dear reviewer 3,
many thanks for reviewing our paper. Please find attached a pdf-File in which we describe how we dealt with your concerns/suggestions.
Regards
Ralf Weiskirchen
